# What Every Diabetologist Should Know about SARS-CoV-2: State of Knowledge at the Beginning of 2021

**DOI:** 10.3390/jcm10051022

**Published:** 2021-03-02

**Authors:** Urszula Abramczyk, Aleksandra Kuzan

**Affiliations:** 1A. Falkiewicz Specialist Hospital in Wroclaw, 52-114 Wroclaw, Poland; u.abramczyk@gmail.com; 2Department of Medical Biochemistry, Wroclaw Medical University, 50-368 Wroclaw, Poland

**Keywords:** diabetes, COVID-19, SARS-COV-2, atherosclerosis

## Abstract

For almost a year, the major medical problem has been the pandemic caused by the SARS-CoV-2 virus. People with diabetes who contract COVID-19 are likely to experience more serious symptoms than patients without diabetes. This article presents new research about the epidemiology of COVID-19 in a group of patients with diabetes. It details the mortality and prognosis in such patients, as well as the relationship between COVID-19 and the diseases most often coexisting with diabetes: obesity, atherosclerosis, hypertension, and increased risk for infection. It also details how the virus infects and affects patients with hyperglycemia. The context of glycation and receptors for advanced glycation products (RAGE) seems to be of particular importance here. We also present a hypothesis related to the cause-and-effect axis—it turns out that diabetes can be both the cause of the more difficult course of COVID-19 and the result of SARS-CoV-2 infection. The last part of this article discusses the impact of antihyperglycemic drugs on the development of COVID-19 and other pharmacological implications, including which non-classical antihyperglycemic drugs seem to be effective in both the treatment of coronavirus infection and glucose homeostasis, and what strategies related to RAGE and glycation should be considered.

## 1. Introduction

Diabetes is one of the biggest risk factors for dying from Coronavirus Disease 2019 (COVID-19), an illness caused by Severe Acute Respiratory Syndrome Coronavirus 2 (SARS-CoV-2). COVID-19 first appeared in 2019, before triggering the pandemic in 2020 [1,2].

According to the International Diabetes Federation, 463 million adults between 20–79 years of age suffer from diabetes (2019) [3]. This data suggest 20–50% of patients with a positive PCR test for SARS-CoV-2 have chronic diabetes [4]. It is the second most common disease associated with the SARS-CoV-2 infection [5], and it affects all age groups. However, COVID-19 is most often diagnosed in people aged 47–59 and rarely in people under 20 years of age [6,7]. It should be noted that this data may be inconclusive as many people below the age of 20 who contract the virus may not show any symptoms [8]. Infections are almost equally common in both sexes, although a slightly higher number of cases are reported for men, which may be due to their higher incidence of chronic cardiovascular disease [6,9]. Current research indicates a higher risk of developing the disease among patients with type 2 diabetes, with speculation on whether the cause is the presence of diabetes or its complications if it is not controlled [7,10]. At the same time, there are scientific reports indicating that diabetes does not increase the risk of SARS-CoV-2 infection, but only worsens the course of respiratory infections caused by this virus [6,11].

## 2. Mortality among People with Diabetes

Compared to patients without diabetes, patients with both COVID-19 and diabetes exhibit higher mortality rates [12,13,14,15]. A study in Scotland on 5,463,300 patients showed that people with diabetes who contracted SARS-COV-2 were more likely to be treated by a lethal or intensive care unit than those without diabetes with the odds ratio being 2.4 for type 1 and 1.4 for type 2 diabetes [13].

The study was done on the English population (61,414,470 people) shows that there is a difference between the probability of hospital death due to SARS-CoV-2 infection in people with type 1 and type 2 diabetes. Patients with type 1 diabetes mellitus (T1DM) had a risk of hospital death 3–5 times higher than patients without diabetes, while patients with type 2 diabetes mellitus (T2DM) were only twice as likely to die than patients without diabetes [15]. This analysis also showed a significant increase in mortality rates in patients with diabetes and in elderly patients. The mean age of death in patients with type 1 diabetes is 72.2 years (SD 13.0), 77.9 in those with type 2 diabetes (SD 11.0), and 79.2 years (SD 12.5) in patients without diabetes. It should be added that the risk of death for people under 40, suffering from type 1 and 2 is pretty low, but still higher when compared to people without diabetes [15]. When interpreting this data, one should bear in mind there are some quite serious limitations. Only three cardiovascular comorbidities were included in the analysis and the analysis did not take into account hypertension, chronic kidney disease, BMI of patients, or non-hospital deaths of patients [15]. According to the research of other authors, the mortality rate due to COVID-19 in people with diabetes is similar for T1DM and T2DM [16].

## 3. Prognosis among People with Diabetes

The available research shows that age is the most significant prognostic factor in patients with COVID-19 and diabetes [2,5,12]. The older the patient is the worse their prognosis will be.

The severity of impairment in glucose control plays an important role in patients with diabetes, regardless of which type they have. Studies have proven that high glucose concentration affects the respiratory epithelium which significantly increases the possibility of infection of the influenza virus, which, like the SARS-CoV-2 virus, affects the respiratory tract [17]. Proper glycemic control may improve the prognosis of SARS-CoV infection [18]. Retrospective studies conducted on a group of 952 people suggest keeping glycemia within the range of 3.9 to 10.0 mmol/L as it reduces mortality [19]. Elevated blood glucose levels affect mortality in both people with and without diabetes [12].

Since blood glucose control is more important than ever before and access to diagnostic laboratories that can determine glycated hemoglobin HbA1c is more difficult, it is suggested to use the latest technology to achieve this goal. The proposed method is real-time continuous glucose monitoring (RT-CGM) [20]. Using modern devices, it is possible to detect nocturnal or unrecognized hypoglycemia and glycemic variability. The disadvantages of this procedure are: lag time between the sensor and blood glucose, the need for calibration, false detection incidents, and mild discomfort or skin irritation reported in some users [21]. However, the benefits outweigh the disadvantages, i.e., accuracy, effectiveness, convenience for the patient, and the possibility of population control during a pandemic [20,21].

Atherosclerosis in patients with diabetes significantly worsens the prognosis of patients with SARS-CoV-2 [22].

Low levels of albumin and high levels of CRP may also be a poor prognostic factor in patients with diabetes and COVID-19 infection. Another still uncertain aspect is the use of insulin in hospitalized patients, as worse treatment outcomes have been obtained in patients with COVID-19 treated with insulin compared to patients treated with other antihyperglycemic drugs [23,24]. It is postulated that insulin treatment promotes systemic inflammation and aggravated injuries of vital organs [23]. On the other hand, analysis of other authors showed that neither insulin nor any other drug was associated with hospital death during SARS-CoV-2 infection [12].

The early normalization of potassium concentration was proposed as a predictor of a good prognosis [25,26,27].

## 4. The Specificity of SARS-CoV-2 Infections in Patients with Different Types of Diabetes

Mortality by major types of diabetes is discussed in Chapter 2. Other parameters are discussed below.

Patients with both type 1 and type 2 diabetes had a greater chance of increased morbidity compared to patients without diabetes, but those with type 2 diabetes have it worse than those with type 1 diabetes while those with type 1 have it worse than those without diabetes [28]. While the possibility of hospitalization and a more severe course of SARS-CoV-2 is almost equally likely in both patients with T1DM and T2DM, they are both 3–4 times more likely to have these possibilities than those without diabetes [15,28]. Interestingly, no significant correlation was observed between the duration of diabetes type 1 and the severity of COVID-19, taking into account age and sex [28].

Attention should also be paid to borderline forms of diabetes, the pathogenesis of which differs from that of diabetes mellitus 1 and 2 described above. In pancreatogenic type 3c diabetes (T3cDM), both insulin resistance and insulin deficiency appear, putting these patients at risk of simultaneous hyperglycemia and hypoglycemia depending on the stage of the disease [29,30]. There are reports proposing that SARS-CoV-2 infection may cause acute pancreatitis [31,32,33], which suggests that recurrent COVID-19 infections could lead to the development of T3cDm, but there aren’t enough studies to support this thesis.

It is reported that the most common complication in pregnant patients with COVID-19 was diabetes when compared to the general population (non-gestational diabetes and gestational diabetes among pregnant patients with COVID-19 were 8% and 10%, respectively) [34]. Euglycemic ketoacidosis has been observed in patients with gestational diabetes who are also infected with SARS-CoV-2 [35].

There is limited data available on the less frequently diagnosed types of diabetes which makes researchers require a more in-depth analysis of patients in this group.

## 5. Diseases Coexisting with Diabetes Affecting the Morbidity and Course of COVID-19 in People with Diabetes

### 5.1. Obesity

Obesity is one of the biggest health problems of our generation. According to WHO estimates, 650 million adults are obese [36]. The available sources show differences in the risk of disease and the course of COVID-19 infection depending on BMI [37,38,39]. It has been established that a BMI above 30 kg/m^2^ predisposes a person to severe COVID-19, and people who have a BMI above 35 kg/m^2^, have a significantly higher risk of dying from SARS-COV-2 [40]. A BMI of over 40 could increase the risk of death by 90% [41]. Regardless of the patient’s age, the frequency of mechanical ventilation increases with the increase in the patient’s BMI, which has also been observed in children [42,43]. Both diabetes and obesity have a significant impact on the development of chronic systemic inflammation which exacerbates the “cytokine storm” arising from COVID-19 infection [26]. It is postulated that the mechanisms underlying the relationship between the worst course of COVID-19 and obesity are universal mechanisms for various infections, including influenza A, mainly higher levels of circulating leptin and pro-inflammatory cytokines, reduced macrophage activation, both T cell and B cell response associated with the obesity-related chronic inflammatory state and increased viral spread to the alveolar region of the lungs [40].

It should also be noted that the lockdown introduced in many regions promoted weight gain and worsened glycemic control [26,44]. Some countries, such as England, developed specific government strategies, such as including the taxation of unhealthy food, to protect citizens from the consequences of obesity during a pandemic [41].

On the other hand, however, it is suggested that there is an “obesity survival paradox” in SARS-CoV-2 infection—as a result of anesthetic inconveniences in obese patients with pneumonia, earlier and more intensive life-saving activities are undertaken compared to patients with normal body weight [45]. It has also been taken into account that lower mortality in obese people may also result from an increased metabolic reserve [45]. The accumulated triglycerides are the energy fuel used in the body’s fight against infection. The amount of accumulated lipids in obese patients probably allows for a better response of the body during increased catabolic stress during pneumonia, compared to people with a normal body weight [46]. Such a mechanism is suspected to occur in SARS-CoV-2 infections, at least for some patients [45].

### 5.2. Atherosclerosis

A damaged endothelium caused by diabetes may predispose a sufferer to more severe complications from COVID-19 infection, as it has been suggested that the virus attacks endothelial cells in a similar manner to respiratory cells using the ACE2 receptor, causing inflammation in these cells [39,44,47]. Another suggested mechanism for inducing vascular lesions during COVID-19 infection is the direct reaction of pro-inflammatory cytokines with atherosclerotic plaque, which may increase the risk of thrombosis due to the damage it takes [48]. This explains such a diverse clinical picture in patients with COVID-19 infection because inflammatory changes affect the endothelium throughout the body [47,49]. In one study, it was hypothesized that the severity of atherosclerosis correlates with the severity of infection, while also suggesting that SARS-CoV-2 infection exacerbates atherosclerosis, creating a vicious cycle [22].

### 5.3. Hypertension

It is known from the studies available today that arterial hypertension significantly increases the risk of severe respiratory diseases [22]. The simultaneous occurrence of diabetes mellitus and arterial hypertension is one of the factors that also exacerbates the course of infection with SARS-CoV-2. The most controversy in the case of hypertension arises occurs when choosing the appropriate treatment, due to reports of the harmful effects of ACEI/ARBs [22,26,27,50,51]. Although. recent studies state that ACEi/ARBs have no adverse effects during COVID-19 treatment [52]. This topic will be developed in more detail in the next part of this article.

### 5.4. Infections Tendency

Inflammation in diabetes is usually associated with excessively high or inadequately controlled blood glucose levels. Diabetes enhances the cytokine response in infections, and studies have shown that people with diabetes had significantly higher levels of IL-6 and other pro-inflammatory cytokines compared to people without diabetes in SARS-CoV-2 infection [26].

People with diabetes are more likely to develop out-of-hospital pneumonia [53], which puts this group of patients at greater risk in the event of a COVID-19 infection and the way the virus spreads. Improper glycemic control may also lead to an increase in other secondary infections [26].

It should be noted that during the hospitalization of patients with SARS-CoV-2, the blood glucose level depends on many factors, including the drugs used, and the course of the infection itself [24]. Its proper control may positively influence susceptibility to infections.

## 6. The Mechanisms Linking Diabetes and COVID-19

### 6.1. Relationship between the Major Receptor for SARS-CoV-2: ACE2 and Diabetes

An essential receptor for viral cell entry is ACE2 (Angiotensin-Converting Enzyme 2). Physiologically ACE2 plays an important role in the Renin–angiotensin–aldosterone system (RAAS) and the imbalance between the ACE/Ang II/AT1R pathway and the ACE2/Ang/Mas receptor pathway in the RAAS system results in blood pressure disturbances and multi-system inflammation [54].

It turns out that ACE2 is also associated with glucose metabolism. It was concluded on the basis of experiments conducted more than a decade ago, in which knock-out of the ACE2 gene in mice resulted in a deficiency of pancreatic insulin secretion that was partially countered by increased glucose utilization in muscle. Furthermore, overexpression of ACE2 improved glycaemic control in a model of type 2 diabetes in mice [55,56]. It is also reported that insulin decreases the expression of ACE2 [57] and levels of ACE2 in urine and serum are increased in patients with diabetes and positively related to blood glucose and HbA1c levels [56]. Since it is found that diabetes may be causally related to increased ACE2 expression in lung tissue, we can conclude that it may increase viral entry facilitating infection with SARS-CoV-2 in the lung [56].

It is worth noting that similar mechanisms also took place in the case of the SARS-CoV virus, which was identified in late 2002 and caused the epidemic in 2003 [58]. It has 79% genetic homology with the SARS-CoV-2 virus and enters cells via the same ACE2 receptor [58]. Similar effects were observed in patients infected with the SARS-CoV virus as in patients of the current pandemic—in addition to the respiratory consequences, a pattern was also seen in people with diabetes [18,58,59].

### 6.2. The Relationship between Glycation Intensified during Diabetes and COVID-19

Glycation is a non-enzymatic process where reducing sugars, including glucose, glucose-6P, and other monosaccharides, oxaldehydes, dicarbonyls, and other derivatives of sugar metabolism, react with the amino groups of lysines or arginines in proteins. Due to the fact that people with diabetes have a higher concentration of the substrate, the process is intensified in such patients. As a result of numerous complex reactions, including the creation of Schiff’s base, Amadori rearrangement, oxidation, dehydration, fragmentation, condensation with other amino groups, the so-called advanced glycation end-products (AGEs) are formed. Proteins modified by glycation have altered mechanical and biochemical properties, e.g., due to the cross-linkage between the molecules, they are rigid, resistant to proteolysis, and not reactive with the relevant ligands. The consequences of these changes are angiopathies, nephropathies, neuropathies, and other diseases accompanying diabetes [60].

Sartore hypothesized that glycation of ACE2 directly affects the pathomechanism of COVID-19 development. The author reported that in a single ACE2 molecule, 34 lysine residues are exposed, and at least one of these is involved in a hydrogen-bond interaction with the SARS-CoV-2 receptor-binding domain (RBD). Glycation of these residues would cause a change in the protein tertiary structure, perhaps this would contribute to the overexpression of the receptor or a change in its activity [61]. However, this is still a hypothesis that requires verification.

Another less direct link between glycation and COVID-19 would be the modification of antibodies that target viral proteins. With high sugar levels, Fab fragments are more prone to be modified by glucose, particularly in the light and heavy variable regions, which reduces the effectiveness of the antibodies, and thus reduces the ability to fight the virus SARS-CoV-2 immunologically [61].

### 6.3. Diabetes Mellitus as a Cause of Micro-Clots and Blood Clots in the Course of COVID-19

As mentioned before, one of the classic consequences of hyperglycemia is the complications in the form of micro- and macroangiopathy. This is related to the impaired function of vascular endothelial cells in patients with diabetes, which is due, inter alia, to the presence of cross-links caused by glycation and as a result of oxidative stress accompanying hyperglycemia. Ackermann et al. analyzing the material from the autopsy showed that alveolar-capillary microthrombi were nine times as prevalent in patients with COVID-19 than in patients with influenza. It was concluded that vascular endothelial dysfunction might be involved in the increased mortality of diabetes patients with COVID-19 [62].

## 7. RAGE and COVID-19

AGEs bind to numerous receptors, including receptors for advanced glycation products (RAGE). When combined with a ligand, they activate numerous pro-inflammatory metabolic pathways, including the activation of NADPH oxidase, which generates ROS, and a Ras-dependent pathway that activates ERK1/2 and p38MAPK kinases, which ultimately leads to the activation of NFkB and expression of adhesion molecule genes, TNFα and pro-inflammatory interleukins (IL-1, IL-6) [1,63].

RAGE is physiologically expressed at low levels in many tissues and increased during inflammation. The exception is alveolar epithelial cells type I and type II (AT1, AT2), where expression is generally high. In the tissue, RAGE seems to be a critical mediator of the inflammatory response. It is reported that activation of the angiotensin II 1 receptor (AT1R) by angiotensin II (AngII) can transactivate RAGE, which stimulates an inflammatory response in the lungs [56]. Infection with SARS-CoV-2 perturbs the renin-angiotensin system causing AT1R activation, which affects RAGE, enhancing its pro-inflammatory and pro-fibrotic effects [56].

### Possible Role of RAGE in SARS-CoV-2 Cell Entry

Among the numerous interactions in which RAGE is involved, it also activates the protein CD147 (also known as Basigin or EMMPRIN). This protein is highly expressed in type II pneumocytes and in other cells, including immune cells, endothelial cells, and platelets. Its expression is unregulated in hyperglycemia. CD147 is involved in hyaluronan production, which probably directly affects the development of blood clots and inflammation in the course of COVID-19 [63,64]. Some publications report that CD147 glycation increases the expression of metalloproteinases, which relaxes the lung tissue and facilitates viral invasion. Other reports say that CD147 is one of the two, apart from ACE2, recognition receptors for Spike protein (SP) of the virus, thus directly enabling invasion. Azithromycin may be a CD147 blocking drug that can inhibit invasion and alleviate COVID-19 progress [64]. Therefore, from a diabetology perspective, the relationship between RAGE and CD147 in the course of COVID-19 appears to be clinically very significant.

Due to the unexplained issue of why some patients infected with SARS-CoV-2 are asymptomatic and some have life-threatening conditions, Rojas emphasizes that RAGE is a highly polymorphic protein and hypothesizes that some polymorphisms may contribute to the development of symptoms of acute respiratory failure in some patients [54]. The scheme of the interaction between coronavirus and the receptors discussed in the text is shown in Figure 1.

## 8. Diabetes as a Result of COVID-19

ACE2, in addition to the cells of the respiratory system, is also expressed in pancreatic endocrine tissues. Fignani et al. proved that ACE2 is highly expressed in subsets of insulin-producing β-cells and pancreas microvasculature pericytes and is moderately expressed in scattered ductal cells and that pro-inflammatory cytokines increase ACE2 expression in the β-cell line (EndoC-βH1) and in primary human pancreatic islets [65]. It is stated that SARS-CoV-2 binds to the ACE2 receptor in islet cells, causing islet cell damage [51,65]. The same was the case with SARS-CoV-1 and other enteroviruses, which use the same receptor as SARS-CoV-2. Some patients infected with SARS-CoV-1 developed transient diabetes [56]. So it seems that diabetes may not only be the cause of the severe course of COVID-19 but also may be a result of this disease. This phenomenon is suspected to be particularly problematic in patients with pre-existing disturbances in glucose homeostasis [56].

## 9. The Role of Vitamin D in the Pathophysiology of COVID-19 in People with Diabetes

Vitamin D’s primary function is to aid the absorption of calcium and phosphorus, but it has a variety of other functions, including supporting the immune system and regulating glucose homeostasis. Focusing on the relationship of vitamin D with diabetes, it should be mentioned that this compound activates pancreatic beta-cells and regulates calcium hemostasis which has a positive effect on insulin secretion and sensitivity [66]. It has been proven that vitamin D deficiency is common in patients with diabetes [66,67,68].

A meta-analysis of Periera et al. reveals that vitamin D deficiency (25(OH)D below 50 nmol/L) has been associated with severe COVID-19 [69]. This is not surprising since we know that vitamin D is involved in the modulation of the innate and acquired immune system [68]. It stimulates lymphocytes to produce cathelicidins and defensins, reducing the survival and replication of viruses and supports maintaining endothelial integrity [70,71]. This vitamin may play a special role in the course of COVID-19, since it can inhibit proinflammatory cytokine production in human monocytes and macrophages, hence counteracting the “cytokine storm”. Chronic vitamin D deficiency may induce RAS activation, leading to the production of fibrotic factors and, as a consequence, lung damage [68,70].

It is suggested to obtain and hold 40–60 ng/mL (100–150 nmol/L) 25-hydroxyvitamin D (25(OH)D) in blood serum for prevention of COVID-19 (in practice, for people with a deficiency, this may mean supplementation with doses 10,000 IU/d of vitamin D_3_ for a few weeks, followed by 5000 IU/d) [71]. For the treatment of people infected with SARS-COV-2, an even higher vitamin D_3_ concentration might be needed. Recommendations are prepared based on randomized controlled trials [71].

## 10. Interactions between the Development of COVID-19 and the Intake of Antihyperglycemic Drugs

### 10.1. Insulin

During COVID-19 infection, pancreatic beta cells are forced to secrete an increased amount of insulin due to infection, which is already difficult in people with diabetes. In the light of recent studies, this may prove to be particularly difficult for the body, because the virus has a direct affinity for ACE receptors also found on pancreatic beta cells, which may lead to uncontrolled hyperglycemia [26,44]. While focusing on treating SARS-COV-2 infection, hypoglycaemic episodes that may arise from typical metabolic disturbances in diabetes should not be ignored [6]. Therefore, adjusting the insulin dose is quite a challenge when treating SARS-CoV-2 infection.

During hospitalization of people with diabetes due to SARS-CoV-2, elevated glucose levels are most often observed, caused by both the infection itself and the use of glucocorticosteroids and antiviral drugs (lopinavir/ritonavir) during treatment [24,72]. The most common treatment regimen is an abrupt switch to insulin. Some authors warn that the solution involves some risk as sudden high variability of glucose concentrations during hospitalization exacerbates inflammatory changes by increasing the oxidative stress in cells [73]. Hypoglycemia may also occur during insulin dose adjustment, with a negative impact on cardiovascular mortality [24]. Some authors maintain, however, that insulin should be still the main approach to the control of acute glycemia, especially for critically ill patients. In the case of patients with COVID-19, one should not avoid insulin therapy despite the fact that it is tied to hypoglycemia risk [74,75].

### 10.2. Metformin

Metformin was initially proposed as a drug in host-directed therapy for COVID-19. One meta-analysis compared the effects of metformin with other hypoglycemic drugs used in the treatment of diabetes and showed that in the group of patients treated with metformin, the mortality of hospitalized patients was lower [76,77]. The data of Crouse et al. show that in patients taking metformin, the mortality in the course of COVID-19 is 11%, which is comparable to the general population, without diabetes, and significantly lower than for people with diabetes taking insulin, where the mortality is 24% [77]. The mechanism of beneficial effects on COVID-19 infections would be associated with the improvement of the immune response, reduction of inflammation, and increased protection of cells against oxidative stress and anti-thrombotic effect [76,77,78]. It is even suggested that metformin should be included in patients without diabetes, as it may play a beneficial role in the proper management of patients [26,76]. However, it is speculated that metformin acts synergistically with ARB/ACEI, which would increase ACE2 expression and thus increase the risk of SARS-CoV-2 infection [79]. Unfortunately, there is still no research to verify this thesis.

Currently, there are no contraindications for the use of metformin in patients with diabetes infected with SARS-CoV-2, provided that metformin ensures proper glycemic control.

### 10.3. Tiazolidinedionas glp-1 Analogues, Gliflozins (Inhibitors of Sodium/Glucose Cotransporter2/SGLT-2)

Pioglitazone is a representative of thiazolidinediones and, in addition to its hypoglycemic effect, it also has an anti-inflammatory effect. When used for 30–45 mg/day for three months, it can significantly reduce the level of pro-inflammatory cytokines in non-diabetic patients with insulin resistance. However, this hypothesis has not yet been confirmed [26,80,81].

Pioglitazone and liraglutide, analogs of glucagon-like peptide 1 (GLP-1), increase the expression of ACE2 [6,27,72]. It was demonstrated, for example, for pioglitazone by Romaní-Pérez in a rat model of lung hypoplasia with streptozotocin-induced diabetes [82] by Zhang in rats with steatohepatitis [83] and for liraglutide by Yang on high-fat-induced NAFLD model in mice [84]. However, scientific evidence to date has not shown any changes in the use of these drugs during COVID-19 infection [6,27,72]. It is worth noting that there are reports on GLP-1 analogs that can prevent the SARS-CoV-2 virus from entering the cell due to competitive binding to ACE2 [26].

The use of empagliflozin (an SGLT-2 inhibitor) significantly reduces the risk of cardiovascular death in both people with and without diabetes, which prompts the use of this drug in people with diabetes infected with SARS-CoV-2 [85]. It is worth remembering that GLP-1 analogs, as well as SGLT-2 inhibitors, also reduce the risk of cardiovascular death, additionally prevent kidney disease and support the reduction of body weight, which in patients with obesity and complications of poorly controlled diabetes may positively affect the course of SARS-CoV-2 infection [26,86].

However, the availability of drugs in a given region and complications of the use of GLP-1 agonists and SGLT-2 inhibitors—in which an increased risk of ketoacidosis in people with diabetes has been noted [26]—should be taken into account and the balance of benefits and risks for the patient should be taken into account.

### 10.4. Angiotensin-Converting-Enzyme Inhibitors and Angiotensin II Receptor Blockers

It is reported that people with diabetes have a reduced expression of the angiotensin-converting-enzyme 2 (ACE2), which is responsible for the conversion of angiotensin I and II, and thus also has an anti-inflammatory and antioxidant role. Disruption of these processes causes patients with diabetes to be at increased risk of acute respiratory failure in the event of COVID-19 infection, due to the inhibition of the above-mentioned mechanisms [26].

ACE2 expression is significantly increased in patients taking ACE inhibitors or ARBs. SARS-CoV-2 uses ACE2 as a receptor to infect the host’s pneumocytes (Figure 1) and, upon entering the cell, lowers the concentration of ACE2 on the pneumocyte surface [27]. Feng et al. have given the hypothesis that the increased expression of ACE2 as a result of ACEI/ARBs would facilitate COVID-19 infection [6,50]. Currently, there is no strong evidence for the above-described mechanism [6,12,26,87]. However, this possibility should be taken into account and blood pressure should be assessed in patients with diabetes, because, due to the down-regulation mechanism, ACE2 may contribute to an increase in blood pressure [26]. However, there are new publications drawing the opposite conclusions. Liu et al. found that in people with severe COVID-19, their concentration of angiotensin II is increased and correlates with the severity of changes in the lungs and the amount of viremia in samples from the lower respiratory tract. Thus, the authors suggested that ACEI/ARB drug therapy may even be one component of COVID-19 adjunctive treatment [88]. Another group of researchers postulates that the increase in the concentration of the circulating form of ACE2 may be beneficial in SARS-CoV-2 infection because by binding to the circulating receptor, the pool of viruses attacking cells via the membrane ACE2 is reduced [89]. Studies comparing the death rate depending on the use of ACEI/ARB and other antihypertensive drugs show that patient mortality among these two groups is lower in patients taking ACEI/ARB [22,90]. Despite preliminary reports on the likely increase in the risk of developing the disease with the use of ACEI/ARB, many international organizations consider the evidence to be insufficient and do not recommend discontinuation of therapy with these drugs [26,72,89].

### 10.5. Hydroxychloroquine

It has been shown that hydroxychloroquine (HCQ) can be successfully used in patients with diabetes due to its hypoglycemic effect [72,91]. The mechanisms by which the drug affects glycemic control are still not entirely clear, however, it is suggested that it improves insulin sensitivity, increases insulin secretion, reduces hepatic insulin clearance, and reduces systemic inflammation [92]. This drug appeared to be effective in treating COVID-19 [72,91]. It has been postulated that it can counteract infection by preventing the cleavage of coronavirus surface spike proteins and the subsequent fusion between the viral envelope and lysosomal or endosomal membrane [93]. Anti-inflammatory properties of the drug, confirmed on rheumatic diseases, should also alleviate the course of the COVID-19 [93,94]. Unfortunately, the latest randomized clinical trials do not confirm the effectiveness of hydroxychloroquine in the prevention or treatment of SARS-CoV-2 infections [93,94]. It is suspected that the cause is a delay in the host antiviral response by HCQ [93]. The Food and Drug Administration (FDA) has recently revoked the Emergency Use Authorization (EUA) for emergency use of HCQ and chloroquine to treat COVID-19 [93].

### 10.6. Tocilizumab

Tocilizumab is a recombinant humanized anti-interleukin-6 receptor monoclonal antibody [95], an immunosuppressive drug, mainly used to treat rheumatoid arthritis and severe arthritis in children. It has been proven that hyperglycemia might be responsible for the overproduction of IL-6 [96], which causes a worse prognosis in COVID-19 patients with improperly controlled glycemia [97].

The latest research (2021) shows that there is no evidence of a clinical benefit on lung function or on length of the hospitalization, associated with the use of tocilizumab in patients infected with COVID-19 [95].

According to Marfell et al., a group of COVID-positive patients with hyperglycemia at the level of IL-6 was higher than in a group with normoglycemia. The same publication proved that elevated IL-6 levels in hyperglycemic patients weaken the effect of Tocilizumab [97]. That is why we believe patients should be encouraged to maintain the proper level of glycemic control because they can most likely reduce the risk of a severe SARS-CoV2 infection. Marfell concludes that Tocilizumab is ineffective during hyperglycemia [97].

Despite these negative results concerning tocilizumab, some report that the advantage of using it is not limited to only the inflammatory response but also in patients with diabetes as it improves insulin resistance and lowers HbA1c [72].

### 10.7. AGE/RAGE Inhibitors

After studying the data on the relationship between COVID-19 and glycation, we believe there is a possibility that AGEs and RAGE may be therapeutic targets. The so-called AGEs blockers (aminoguanidine, LR-90), AGEs breakers (e.g., ALT7-11, TRC4186), and anti-RAGE antibodies or RAGE inhibitors blocking its interaction with the ligand (e.g., azeliragon) are worth considering [63]. It has been suggested that blocking the RAGE receptor can significantly inhibit cytokine storm and the thrombotic manifestations in COVID-19, which is of particular importance in patients who overexpress RAGE due to high AGE levels, i.e., in patients with diabetes [63]. Inhibiting the formation of glycation products would block the metabolic pathway associated with RAGE even earlier, and breaking the cross-links arising during glycation would reduce the effects of glycation associated with tissue stiffness and loss of their functionality [98]. Unfortunately, they are associated with some hepatotoxicity and various adverse effects [99], which is why for many years there have been no optimistic reports in the research on new drugs that inhibit or counteract glycation.

## 11. Vaccination against COVID-19 in People with Diabetes

After all the required clinical tests, at the end of 2020, the first vaccination of European citizens with a vaccine based on the mRNA of the Spike protein began. In the first half of 2021 four are already available: two mRNA-based vaccines (Pfizer-BioNTech and NIH-Moderna) as well as two non-replicating viral vector-based vaccines (Oxford-AstraZeneca and Janssen) [100]. It should be noted that vaccines have high effectiveness, over a 90%, in protecting against severe COVID-19 [101,102]. It was found that the humoral response to SARS-CoV-2 in patients with diabetes was present and statistically the same in terms of time to antibody appearance and titers as in patients without diabetes. Therefore, no difference in response to the vaccine is expected in people with or without diabetes [103]. It is emphasized that people with diabetes are at increased risk of COVID-19 and should be given priority access to vaccinations [16,104]. Based on epidemiological studies, it is postulated that patients with type 1 and type 2 diabetes should be treated equally, without distinguishing one of the types as more at risk of severe course and death due to COVID-19 [16].

## 12. Summary

COVID-19 is a new disease (at the time of writing this manuscript, it is only nearly a year since the first case was found) and we are now collecting data on epidemiology and analyzing data on the potential pathomechanism of this disease. Research on a larger scale is just beginning [51], and we are now relying on a fairly small scale. What we do know for sure is that people with diabetes, especially in old age, have a significantly increased probability of hospitalization, serious complications, and death from SARS-CoV-2 infection [5,12,14,15]. When drawing practical conclusions from the currently available data, it should be emphasized that people with diabetes during a pandemic should control their glucose levels and generally put emphasis on prevention. rtCGM is considered a tool in controlling glucose levels [20]. To regulate glucose levels in people with diabetes infected with SARS-COV-2 insulin and/or metformin are used. As an auxiliary, we also have at our disposal angiotensin-converting-enzyme inhibitors and angiotensin II receptor blockers, inhibitors of SGLT-2 like pioglitazone and liraglutide, anti-interleukin-6 receptor monoclonal antibody tocilizumab, and hydroxychloroquine [4,26,72]. Unfortunately, each treatment has certain limitations and limited effectiveness, hence the still disturbing mortality statistics among people with diabetes infected with SARS-CoV-2. We hope that the vaccinations that are being introduced [100] will significantly reduce the incidence of COVID-19 among people with diabetes and lead to a quick end to the pandemic.

## Figures and Tables

**Figure 1 jcm-10-01022-f001:**
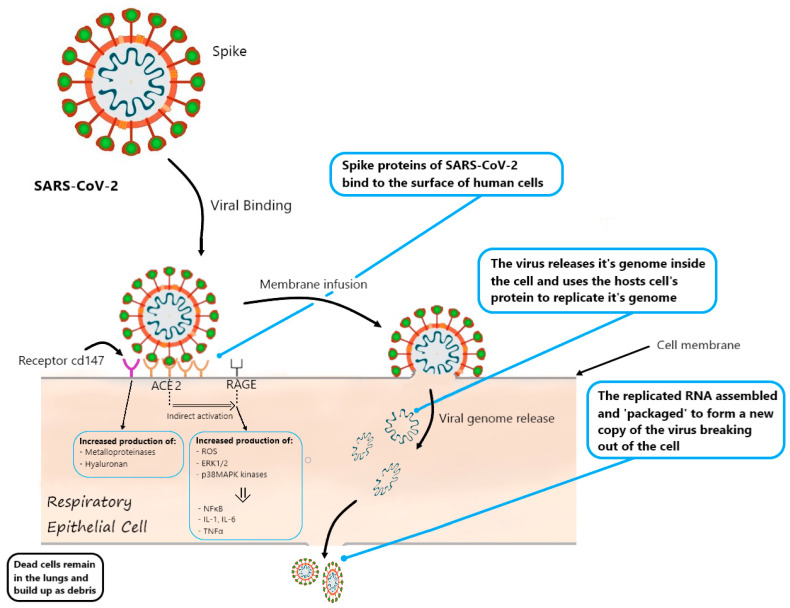
Diagram of infection and its consequences in the epithelial cell of the respiratory system.

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
