# Peer review of "What Every Diabetologist Should Know about SARS-CoV-2: State of Knowledge at the Beginning of 2021"

_jcm, 2021, doi:10.3390/jcm10051022_

Round 1

Reviewer 1 Report

This manuscript aims to review the impact that diabetes has on severity and mortality associated with SARS-COV2 infection, the comorbid factors that increase risk of severe illness in people with diabetes and the possible mechanisms linking increased risk of severe COVID-19 in diabetes.

The manuscript aims to be comprehensive with topical content but in doing so has left out references to literature that has been published on COVID-19 and diabetes. Given that the literature available review is fairly small in scope, review and reference to all published papers should be comprehensive. As an example, the Gregory et al. paper published in December of 2020 is a prospective study looking at COVID-19 and diabetes, which most definitely should be included in this review. It is also important to clarify the differences between T1D, T2D and other forms of diabetes from what has been published. Having more data and analysis included from key papers would add to the strength of this review.

Following are some specific comments:

Line 32-33:  Should this read, “The data available in the literature show that 20-50% patients with a positive PCR test for SARS-CoV-2 have chronic diabetes.”?

Line 35-36: This is confusing as COVID affects all ages similarly but the severity is different between ag groups which skews some of the data if you are only looking at hospitalizations, etc. as to prevalence amount all age groups. It is actually most prevalent according to CDC in young adults. There needs to be more clear delineation throughout the manuscript of risk of infection vs risk of severity/mortality.

Line 49-58: Critical analysis of the UK study is very important here. This study did not clarify that additional risk factors like heart disease were of great impact to outcomes, and that the study actually showed that people with Type 1 diabetes and no other underlying risk factors like older age or other health history actually did quite well – they were not frequently hospitalized for COVID-19 and those who were had low frequencies of severe outcomes. 

Line 152: Important to discuss hyperglycemia that occurred with SARS-CoV that has 79% genetic homology to SARS-CoV2. You do this later in the paper, but these sections may be able to be combined.

Author Response

We would like to thank the Reviewer for all accurate and numerous comments. We thank that the Reviewer devoted the time to our publication and suggested changes that which have greatly enriched our manuscript. To make the manuscript even more up-to-date, we have added a new chapter on vaccinations that started in 2020. We also added a chapter on the role of vitamin D deficiency in diabetics in the context of COVID-19 as well as the reviewer's requested chapter on the differences between T1DM, T2DM, and other forms of diabetes that we found in the literature. We also removed any typographical errors we found and the text was checked by a native speaker. We hope the Reviewer will be satisfied with our corrections and responses to the review.

We quote the review and provide answers to each comment.

The manuscript aims to be comprehensive with topical content but in doing so has left out references to literature that has been published on COVID-19 and diabetes. Given that the literature available review is fairly small in scope, review and reference to all published papers should be comprehensive. As an example, the Gregory et al. paper published in December of 2020 is a prospective study looking at COVID-19 and diabetes, which most definitely should be included in this review. It is also important to clarify the differences between T1D, T2D and other forms of diabetes from what has been published. Having more data and analysis included from key papers would add to the strength of this review.

  • We have added an entire chapter on "The specificity of SARS-CoV-2 infections in patients with different types of diabetes", including the paper by Gregory et al. Thanks to the Reviewer's suggestions, we enriched the manuscript with many other details and new information. The number of cited publikactions increased from 59 to 105, including a large part from January 2021.

Line 32-33:  Should this read, “The data available in the literature show that 20-50% patients with a positive PCR test for SARS-CoV-2 have chronic diabetes.”? 

  • Yes, that is how this sentence should read correctly, thank you for the correction

Line 35-36: This is confusing as COVID affects all ages similarly but the severity is different between ag groups which skews some of the data if you are only looking at hospitalizations, etc. as to prevalence amount all age groups. It is actually most prevalent according to CDC in young adults. There needs to be more clear delineation throughout the manuscript of risk of infection vs risk of severity/mortality.

  • Here we wanted to quote a paper that stated literally that "2% of those infected were younger than 20 years old"; we have changed this statement to be more precise and to emphasize that COVID-19 affects all ages similarly but the severity is different between age groups

Line 49-58: Critical analysis of the UK study is very important here. This study did not clarify that additional risk factors like heart disease were of great impact to outcomes, and that the study actually showed that people with Type 1 diabetes and no other underlying risk factors like older age or other health history actually did quite well – they were not frequently hospitalized for COVID-19 and those who were had low frequencies of severe outcomes. 

  • Thank you for bringing to our attention the admittedly important limitations of this analysis. We have highlighted them in the manuscript.

Line 152: Important to discuss hyperglycemia that occurred with SARS-CoV that has 79% genetic homology to SARS-CoV2. You do this later in the paper, but these sections may be able to be combined.

  • We have added a section on the genetic homology and functional analogy between SARS-CoV and SARS-CoV-2 virus

We are very grateful for your help in improving our manuscript. We hope that after revisions, the Reviewer will find the manuscript valuable and suitable for publication in Journal of Clinical Medicine.

Yours faithfully

Authors

Reviewer 2 Report

The manuscript by Urszula Abramczyk and Aleksandra Kuzan is a narrative and chatty review article discussing different aspects of COVID-19 pathophysiology in patients with diabetes.

Major  comment

The article has several shortcomings that prevent its potential publication in Journal of Clinical Medicine in its actual form. The text is full of typographical errors and English quality should be extensively improved by a native speaker. Authors fail to cite relevant and updated reference regarding the field. In different paragraphs, it is difficult to follow the manuscript, which seems more a mixture of information merged together rather than a document with a clear outline.

Specific comments

Line 13: amend “ infection-prone” into “increased risk for infection”

Line 21:  both the development of coronavirus infection: do authors mean “prevention of”

Lines 27-30: COVID-19 was first described in China at the end of 2019, and caused the pandemic on March 2020; therefore, the adjective “several” is not proper in this context

Lines 35-36: does this statement refer to COVID-19 or diabetes? Clarify

Line 46: “die more often” amend into “exhibit higher mortality rates”

Line 58: “but still higher than for people” amend into “but still higher as compared to people…”

Line 65: “The level of glycaemia” should be “The severity of impairment in glucose control”

Lines 78-79: “…with other hypoglycemic drugs.” Provide a reference

Lines 78-79: hypoglycemic drugs” should be “antihyperglycemic drugs” throughout the text (abstract included)

Lines 75-81: insulin therapy has been associated with increased mortality; discuss and quote PMID: 33248471

- Obesity paragraph: obesity is now a well-established risk factor for adverse outcomes of COVID-19; authors should outline more this aspect rather than a postulated “obesity paradox”. Discuss and quote: PMID: 32674071 ; PMID : 32718928

- Line 101: please, give more details on the metabolic reserve hypothesis

- Lines 119-120: ACEi/ARBs have no harmul effects according to recent reports published in Lancet (Morales 2020: https://doi.org/10.1016/S2589-7500(20)30289-2 )

-Line 230 paragraph: Authors fail to quote relevant literature here. PMID: 33281748

- Authors should discuss the emerging detrimental role of vitamin D deficiency in  COVID-19 pathophysiology, given the high prevalence of vitamin D deficiency in patients with diabetes (PMID: DOI: 10.3390/nu12113361 ; DOI: 10.3390/nu12040988 ; PMID: 33146028; DOI: https://doi.org/10.1210/clinem/dgaa733 ; DOI: 10.1080/07315724.2021.1877580 ; PMID: 31762905 PMID: 24490013)

- Lines 247-248: “At the same time, one should not ignore the episodes of hypoglycemia that may result from typical metabolic disorders in diabetes”: this is not clear, please rephrase

- Lines 253-255: insulin therapy remains the most effective therapy for inpatient hyperglycemia, especially for critically ill patients according to all the international guidelines; the fact that it is tied to hypoglycemia risk does not mean that one should avoid insulin therapy for that reason

- Metformin paragraph: authors should discuss and cite PMID: 33519709

- Line 280: “without symptoms of hyperglycaemia”: do authors mean in “non-diabetic patients with insulin resistance”

- Lines 282-284:Authors introduce GLP_1RA while still talking about pioglitazone; then they refer generically to cells (type?) in animal models (type?)

- Line 287:  “empaglyphysin” change in  “ empagliflozin “

- Lines 287-290: this sentence is really hard to follow

- Line 302: in this paragraph authors mix the role of ACE2 in COVID-19 pathogenesis and its role on blood pressure. Moreover, they fail to cite recent studies showing that there is no association between ACE-I /ARBs and outcomes of COVID-19; this is a very initial hypothesis that has not been confirmed.

- Line 323 paragraph: authors should discuss more the effects of hydroxychloroquine on glucose homeostasis, specifying, however, that its use has proven to be not effective in COVID-19 prophylaxis and treatment (see and quote PMID: 32401405 ;  https://doi.org/10.1016/S2665-9913(20)30390-8 ; PMID: 32693652  )

- Figure 2 is really hard to follow; moreover, some abbreviations are incorrect (e.g. interferon should be IFN) and are not listed in the caption and neither mentioned in the text;  “agitated macrophage” is an awkward term; from the figure, it seems that tocilizumab blocks IL-6, while it actually blocks IL-6 receptor

- Line 361-362: rtCGM should be discussed in the text in order to be mentioned in the conclusion

- Lines 362-369: these are too strong statement based on speculative or observational data

Author Response

We would like to thank the Reviewer for all accurate and numerous comments. After reading them, it can be concluded that the Reviewer is an expert on COVD-19 and follows the latest publications and reports. We thank that the Reviewer devoted the time to our publication and suggested changes that we tried to take into account with the greatest care. To make the manuscript even more up-to-date, we have added a new chapter on vaccinations that started in 2020. We hope the reviewer will be satisfied with our corrections and responses to the review.

We quote the review and provide answers to each comment.

Major  comment The article has several shortcomings that prevent its potential publication in Journal of Clinical Medicine in its actual form. The text is full of typographical errors and English quality should be extensively improved by a native speaker. Authors fail to cite relevant and updated reference regarding the field. In different paragraphs, it is difficult to follow the manuscript, which seems more a mixture of information merged together rather than a document with a clear outline.

  • Thanks to the Reviewer's suggestions, we enriched the manuscript with many details and new information. The number of cited publikactions increased from 59 to 105, including a large part from January 2021. We also removed any typographical errors we found and the text was checked by a native speaker. We believe that the text is now more coherent, harmonious and will appeal to the Reviewer.

Specific comments

Line 13: amend “ infection-prone” into “increased risk for infectionrt”  à We have changed this wording

Line 21:  both the development of coronavirus infection: do authors mean “prevention of” à not prevention, but treatment, we have changed this wording

Lines 27-30: COVID-19 was first described in China at the end of 2019, and caused the pandemic on March 2020; therefore, the adjective “several” is not proper in this context à we refined this data

Lines 35-36: does this statement refer to COVID-19 or diabetes? Clarify à we wanted to write a note that only 2% of those COVID-19 infected were younger than 20 years old, We have changed this wording because it was extremely imprecise

Line 46: “die more often” amend into “exhibit higher mortality rates” à We have changed this wording

Line 58: “but still higher than for people” amend into “but still higher as compared to people…” à We have changed this wording

Line 65: “The level of glycaemia” should be “The severity of impairment in glucose control” à We have changed this wording

Lines 78-79: “…with other hypoglycemic drugs.” Provide a reference à we added relevant references

Lines 78-79: hypoglycemic drugs” should be “antihyperglycemic drugs” throughout the text (abstract included) à We have changed this wording

Lines 75-81: insulin therapy has been associated with increased mortality; discuss and quote PMID: 33248471 à we have read and cited this publication

- Obesity paragraph: obesity is now a well-established risk factor for adverse outcomes of COVID-19; authors should outline more this aspect rather than a postulated “obesity paradox”. Discuss and quote: PMID: 32674071 ; PMID : 32718928 à we got acquainted with these publications and cited them

- Line 101: please, give more details on the metabolic reserve hypothesis à We developed this hypothesis

- Lines 119-120: ACEi/ARBs have no harmul effects according to recent reports published in Lancet (Morales 2020: https://doi.org/10.1016/S2589-7500(20)30289-2 ) à We have read and cited this publication

-Line 230 paragraph: Authors fail to quote relevant literature here. PMID: 33281748 à We have read and cited this publication

- Authors should discuss the emerging detrimental role of vitamin D deficiency in  COVID-19 pathophysiology, given the high prevalence of vitamin D deficiency in patients with diabetes (PMID: DOI: 10.3390/nu12113361 ; DOI: 10.3390/nu12040988 ; PMID: 33146028; DOI: https://doi.org/10.1210/clinem/dgaa733 ; DOI: 10.1080/07315724.2021.1877580 ; PMID: 31762905 PMID: 24490013) à We've added a whole new section on the relationship between Vitamin D Deficiency and COVID-19 in diabetics; we looked at the suggested publications and cited them, unfortunately apart from DOI: 10.1080 / 07315724.2021.1877580, which we could not find on the Internet

- Lines 247-248: “At the same time, one should not ignore the episodes of hypoglycemia that may result from typical metabolic disorders in diabetes”: this is not clear, please rephrase  à We have changed this wording, we hope it is now understandable

- Lines 253-255: insulin therapy remains the most effective therapy for inpatient hyperglycemia, especially for critically ill patients according to all the international guidelines; the fact that it is tied to hypoglycemia risk does not mean that one should avoid insulin therapy for that reason à We have expanded this paragraph and drawn the appropriate conclusions

- Metformin paragraph: authors should discuss and cite PMID: 33519709 à We have read and cited this publication

- Line 280: “without symptoms of hyperglycaemia”: do authors mean in “non-diabetic patients with insulin resistance” à Yes, sorry for being imprecise, we have improved this wording

- Lines 282-284:Authors introduce GLP_1RA while still talking about pioglitazone; then they refer generically to cells (type?) in animal models (type?) à We have added the details requested by the Reviewer

- Line 287:  “empaglyphysin” change in  “ empagliflozin “ à We have changed this wording

- Lines 287-290: this sentence is really hard to follow à We have changed this wording, we hope it is now understandable

- Line 302: in this paragraph authors mix the role of ACE2 in COVID-19 pathogenesis and its role on blood pressure. Moreover, they fail to cite recent studies showing that there is no association between ACE-I /ARBs and outcomes of COVID-19; this is a very initial hypothesis that has not been confirmed à  we extended this section by adding new data, emphasizing that ACEi / ARBs are still recommended also in diabetics

- Line 323 paragraph: authors should discuss more the effects of hydroxychloroquine on glucose homeostasis, specifying, however, that its use has proven to be not effective in COVID-19 prophylaxis and treatment (see and quote PMID: 32401405 ;  https://doi.org/10.1016/S2665-9913(20)30390-8 ; PMID: 32693652  ) à we reviewed the proposed literature and attached their main theses to the manuscript

- Figure 2 is really hard to follow; moreover, some abbreviations are incorrect (e.g. interferon should be IFN) and are not listed in the caption and neither mentioned in the text;  “agitated macrophage” is an awkward term; from the figure, it seems that tocilizumab blocks IL-6, while it actually blocks IL-6 receptor à  We decided to delete figure 2

- Line 361-362: rtCGM should be discussed in the text in order to be mentioned in the conclusion à We added a few sentences about rtCGM in Chapter 3

- Lines 362-369: these are too strong statement based on speculative or observational data à We removed the controversial and uncertain statements from the summary

Once again, we would like to thank the Reviewer for his comments and suggestions, hoping that our responses will be satisfactory and that the manuscript will be valuable and relevant for publication in the Journal of Clinical Medicine.

Yours faithfully

Authors

Round 2

Reviewer 1 Report

Your revisions have resulted in an improved manuscript. The content of the manuscript is more comprehensive and includes analysis of the available data, which is important. Having another native English speaker proofread your manuscript for readability will be important.

The term "diabetic" should be replaced with "people with diabetes".

Author Response

Thank you for the positive opinion of our article.

As suggested, the term "diabetic" was replaced with "people with diabetes" and another native English speaker proofread manuscript for readability. We hope that the Reviewer will find the current version of the manuscript flawless. Thank you very much once again for all your comments and suggestions that allowed us to improve the manuscript.

Yours faithfully,

Authors

Reviewer 2 Report

The Authors addressed adequately my comments and can be accepted for publication.

Author Response

Thank you very much for stating that the manuscript can be published. The work has been re-checked in terms of language. We believe that in its current version, this manuscript will be valuable literature for readers of the Journal of Clinical Medicine. Thank you once again for your great contribution to improving the manuscript and for the time spent analyzing it.

Yours faithfully,

Authors